An amalgamation of YOLOv4 and XGBoost for next-gen smart traffic management system

Dave Pritul 1 pritul.dave@gmail.com
http://orcid.org/0000-0003-1698-5003 Chandarana Arjun 1
Goel Parth 1
http://orcid.org/0000-0002-9993-9901 Ganatra Amit 2
1 Computer Science & Engineering Department, Devang Patel Institute of Advance Technology and Research, Charotar University of Science and Technology (CHARUSAT) , CHARUSAT Campus, Changa, Gujarat , India
2 Computer Engineering Department, Devang Patel Institute of Advance Technology and Research, Charotar University of Science and Technology (CHARUSAT) , CHARUSAT Campus, Changa, Gujarat , India
Zhang Li
Electronic publication date: 2021 Jun 18
Publication date: 2021
Volume: 7
Electronic Location ID: e586
Received 2020 Nov 11; Accepted 2021 May 18
Copyright: © 2021 Dave et al.
Copyright year: 2021
Copyright holder: Dave et al.
License: This is an open access article distributed under the terms of the Creative Commons Attribution License, which permits unrestricted use, distribution, reproduction and adaptation in any medium and for any purpose provided that it is properly attributed. For attribution, the original author(s), title, publication source (PeerJ Computer Science) and either DOI or URL of the article must be cited.
License URL: https://creativecommons.org/licenses/by/4.0/

Keywords: Object detection, YOLOv4, Machine learning, Deep learning, Computer vision, Regression analysis, eXtreme Gradient Boosting (XGBoost)

Funding: The authors received no funding for this work.

==============================
The traffic congestion and the rise in the number of vehicles have become a grievous issue, and it is focused worldwide. One of the issues with traffic management is that the traffic light’s timer is not dynamic. As a result, one has to remain longer even if there are no or fewer vehicles, on a roadway, causing unnecessary waiting time, fuel consumption and leads to pollution. Prior work on smart traffic management systems repurposes the use of Internet of things, Time Series Forecasting, and Digital Image Processing. Computer Vision-based smart traffic management is an emerging area of research. Therefore a real-time traffic light optimization algorithm that uses Machine Learning and Deep Learning Techniques to predict the optimal time required by the vehicles to clear the lane is presented. This article concentrates on a two-step approach. The first step is to obtain the count of the independent category of the class of vehicles. For this, the You Only Look Once version 4 (YOLOv4) object detection technique is employed. In the second step, an ensemble technique named eXtreme Gradient Boosting (XGBoost) for predicting the optimal time of the green light window is implemented. Furthermore, the different implemented versions of YOLO and different prediction algorithms are compared with the proposed approach. The experimental analysis signifies that YOLOv4 with the XGBoost algorithm produces the most precise outcomes with a balance of accuracy and inference time. The proposed approach elegantly reduces an average of 32.3% of waiting time with usual traffic on the road.

Introduction

Traffic management and pollution arising from the traffic are significant issues in India’s metropolitan areas, resulting in unnecessary waiting time and congestion. The transportation is the third predominant cause of air pollution, according to the official data of Environmental Statistics 2019 (Indian Ministry Of Statistics, 2019). Although India’s government began to encourage public transport and escalated the private vehicle taxes, the effect is minimal. In recent times different methods for the regulation and management of the traffic, primarily Internet of Things (Greengard, 2015), Time Series Forecasting (Jenkins, 1994) and Digital Image processing (Gonzalez & Wooda, 2006) have been proposed. Although these approaches achieve impressive results, they are still not cost-effective, fast, or accurate, and they are far from satisfactory in real-time scenarios. To overcome these issues, a method using the Convolutional Neural Network (KrizhevskyA, 2012) is proposed. The advancements and development of the effective algorithms in Computer Vision (Szeliski, 2010) enables the computers to perform the task without the use of sensors or human intervention and convey real-time information. As a result, an Artificial Intelligence-based (Long et al., 2020) Computer Vision solution is proposed to detect the traffic density on the lanes and anticipate the optimum time required to clear the traffic. Moreover, the computer vision-based approaches serves the visual interpretation of the scenarios, resulting in model transparency. The analyzed results validates that the proposed approach is faster at inference, cost-effective, accurate, and operates over CPU (Central Processing Unit) with minimal processing power.

At the intersection, regardless of the number of vehicles present over the lane, the traffic lights timer is in the constant round-robin phase. For instance, during the peak hours, the stream of traffic towards the south is significant, but there is decidedly less traffic flow in the other direction. However, the timer allocated to each lane remains constant, and as a repercussion, extraneous waiting time emerges. The Indian government has already established the CCTV (Closed-Circuit Television) cameras at the intersections and began using electronic format-based paperwork commonly known as e-challan, having a beneficial impact, but yet the dynamic traffic light not implemented. Additionally, in certain areas, the traffic police regulate traffic through hand-held traffic movements, which is arduous and cumbersome. Therefore, a simple and effective solution is proposed to change the timer of the green light window based on the different classes of vehicles present over the lanes. Moreover, the precipitation details are also taken into consideration, as it broadly affects the lane clearance time. The state of the art You Only Look Once (YOLOv4) (Bochkovskiy, Wang & Liao, 2020) algorithm is employed for detecting the vehicles and counting different classes of the vehicles since it is one stage detector and has higher accuracy along with lower inference time. The robust ensemble technique, eXtreme Gradient Boosting Algorithm (XGBoost) (Chen & Guestrin, 2016) is proposed for predicting the optimal time of the green light window as it is fast, efficient, accurate, and prone to overfit. The prediction model is constructed based on analyzing the traffic patterns from the city of Vadodara during rush hours. Furthermore, the Microsoft Common Object in Context (MS COCO) (Lin et al., 2014) dataset is being considered for detecting the different classes of vehicles. After constructing and fine-tuning the prediction and the detection model, the experimental results shows a reduction of 32.3% in the average waiting time of vehicles during the time interval of regular traffic.

Moreover, in this article, different objection detection algorithms and regression-based prediction models are explored. Among them YOLOv4 (You Look Only Once version 4) (Bochkovskiy, Wang & Liao, 2020) and XGBoost (Extreme Gradient Boosting) (Chen & Guestrin, 2016) outperformed in all the constrained scenarios.

The remainder of the article is organized as follows: “Literature Review” presents the related work addressing the Smart Traffic Management framework using different techniques. “Materials and Methodologies” describes the proposed approach along with an algorithm for building the system’s architecture. The models’ results are discussed in “Results and Discussion”, additionally, the models’ output is addressed. “Conclusion” concludes the article by discussing possible future work for smart traffic management systems.

Literature review

Preliminary approaches for smart traffic management systems are IoT-based approaches (Greengard, 2015), Time-Series based approaches (Jenkins, 1994), and Computer Vision-based approaches (Gonzalez & Wooda, 2006). Although these approaches yield plausible results, they are expensive, not repurposing, less trustworthy, and less interpretable.

IoT based approach

Kalaiselvi, Sangavi & Dhivya (2017), Sharma et al. (2018) have proposed a technology-driven solution, Light Fidelity (LiFi), that transmits signals to the traffic control system signaling an emergency vehicle’s arrival. Akhil & Parvatha (2017) discussed the Sound Navigation and Ranging (SONAR) technique which measures the density of vehicles with an array of UltraSonic Sensors. The timer of the green light and red light is derived from SONAR readings. Bui, Jung & Camacho (2017) proposed an IoT-based sensor network combined with the game theory. Every IoT-based entities such as vehicles, sensors, and traffic lights exchange the information in the proposed approach. The Cournot competition model for non-priority vehicles and the Stackelberg competition model for priority vehicles are adopted. Based on this game model, the required time to allocate for a traffic light is determined. Atta et al. (2020) has discussed methodology using RFID sensor modules for sensing the density of vehicles and minimizing the congestion. For every incoming vehicle, the signal is sent to the RFID receiver, and accordingly, the count of vehicles is incremented. Following that, the fuzzy inference is drawn to predict the estimated time. Zambrano-Martinez et al. (2019) proposed a load balancing algorithm in which traffic is routed to the particular lane where there is lesser traffic. A route server is implemented to handle all of the city’s traffic. It includes the SUMO and DFROUTER tools, which produce data from real-world traffic traces. The ABATIS connection interface connects two simulators, the SUMO traffic, and the OMNET++ network simulator. Furthermore, the results are validated by injecting 34,065 vehicles. An 8% increase in travel time and a 16% improvement under heavy loads is observed in their proposed approach.

Time-series forecasting

The time-series methods are based on historical data. Natafgi et al. (2018) proposed an adaptive reinforcement learning approach. In which the multiple agents are assigned to a crossroad and learn optimal time and distance to travel based on reward and penalty of their actions. Ata et al. (2019) proposed Artificial Backpropagation Neural Network for the Smart Road Traffic Congestion, which predicts time delay based on traffic speed, humidity, wind speed, and air temperature. Furthermore, Chen, Chen & Hsiungy (2016) refined this method using Genetic Algorithms. The number of vehicles heading towards the green light and the vehicles halted at the red light are used as parameters. However, historical data is less accurate when compared with real-time scenarios and can yield erroneous results.

Computer vision and machine learning based approach

In the Digital Image Processing-based approach, the image is subtracted using foreground-background subtraction with the reference image, and the blob of the object is obtained. Thereafter the traffic light is controlled by counting the number of detected blobs (Choudekar, Banerjee & Muju, 2011; Frank, Khamis Al Aamri & Zayegh, 2019). However, conventional image processing approaches are not much robust in terms of the changing light conditions, and become skewed against dense traffic. Other proposed methods involve detecting and tracking the rearmost vehicle in the frame (Asha & Narasimhadhan, 2018). However, since determining the last vehicle is difficult, this approach is limited to a local solution. Castaño et al. (2017) and Castaño et al. (2018) introduced the obstacle detection method where the object is recognized using Support Vector Machine. Collision avoidance is accomplished using a Multi-Layer Perceptron and a Self-Organizing Map. Along with that for collision detection, the 3DLiDAR technique is presented and minimized using Reinforcement Learning. In order to recognize objects, the Support vector machine (SVM) classifier is used. Nonetheless, the complexity of SVM escalates as the training sample size increases, necessitating further computation and resulting in overfitting (Caruana & Niculescu-Mizil, 2006). However, Convolutional Neural Networks (CNN) are more advanced than Multi-Layer Perceptrons because filters and kernels extract only the essential features from the image. Harrou, Zeroual & Sun (2020) proposed a piecewise linear traffic (PWSL) and Kalman filter as a virtual sensor for estimating traffic densities. The residuals from the actual and virtual sensors are fed as an input to the unsupervised K-nearest neighbor (KNN) algorithm. The temporal clustering optical flow features based on Temporal Unknown Incremental Clustering are proposed by Kumaran et al. (2019). In that approach, the moving cluster of objects are tracked based on the region of interest. These moving clusters determines the density of vehicles on the lane. The signal time is estimated based on the type of vehicle rather than the number of vehicles. Throughput and Average Waiting Time Optimization Model, which is based on Gaussian regression, has been trained for the departure and arrival rates of different clusters. Lv et al. (2014) predicted the density of traffic at different roads. The 15,000 data points are gathered and analyzed from different sensors. The stacked autoencoder is used to make the prediction. It consists of multiple autoencoders stacked together followed by a logistic regression layer at the top. Furthermore, the results of stacked autoencoders are compared with different neural networks.

This article focuses on efficiently reducing excessive waiting time in real-time scenarios by using computer vision-based object detection and machine learning-based prediction model. As a result, two distinct studies are proposed and discussed in subsequent sections.

Overview object detection algorithms

Prior to advancing the deep Convolutional Neural Network (KrizhevskyA, 2012), Image Processing-based methods for object detections were used. The most widely used methods were Scale-Invariant Feature Transformation (SIFT) (Lowe, 2004), and the Histogram of Oriented Gradients (HOG) method (Dalal & Triggs, 2005). Thereafter, a sliding window approach is introduced for object detection. Region-based Convolutional Neural Network (R-CNN) (Girshick et al., 2014) modified the approach by proposing the “Region Proposals,” which consist of obtaining a subset of the image and then classifying the object using Convolutional Neural Network (Fukushima, Miyake & Ito, 1983). Hence R-CNN is a two-stage detector, as it requires more time for the inference.

Whereas, You Only Look Once (YOLO) (Redmon & Farhadi, 2018) is a one-stage detector in which the image is divided into the SxS grids, each of which acts as a classifier cell predicting the bounding box and confidence ratio. The Fast R-CNN (Girshick, 2015), a variant of R-CNN, has an mAP of 70% with 0.5 FPS, while the YOLO-based model has 63.4% mAP with 155 FPS on Pascal VOC 2007 dataset (Everingham et al., 2012). The mAP value is the mean of average precision, which combines the value of precision and recall values given as ∑rP@rR (Zhang & Zhang, 2009).

YOLOv2 (Redmon & Farhadi, 2017) employs the Batch Normalization, which improved accuracy by 2%. Furthermore, the principle of anchor box prediction is introduced. The bounding box’s dimensions are predicted using KMeans Clustering. The mAP value of YOLOv2 is 76.8%, which is 3.6% higher than Faster RCNN over Pascal VOC 2007 dataset (Everingham et al., 2010).

YOLOv3 improved this loss function, which is exhibited in Eq. (1). The whole loss function of YOLOv3 is formulated on regression loss, confidence loss, and classification loss. Moreover, the concept of different layers for the different sizes of the object is introduced. In Equation, 1 1i jobj states that the output value will be 1 if the box and cell value matches; otherwise, it will be 0. When there is no entity 1i jnoobj reverses the output value.

(1) λcoord∑i=0S2∑j=0B1ijobj(xi−x^i)2+(yi−y^i)2+λcoord∑i=0S2∑j=0B1ijobj(wi−w^i)2+(hi−h^i)2+∑i=0S2∑j=0B1ijobj(Ci−C^i)2+λnoobj∑i=0S2∑j=0B1ijnoobj(Ci−C^i)2+∑i=0S21iobj∑cl∈dasses(pi(c)−p^i(c))2

The YOLOv4 achieves an mAP value of 43% with 43 FPS, whereas FasterRCNN achieves 39.8% mAP with 9.4 FPS over Titan X Pascal GPU.

The Paddle-Paddle YOLO (PP-YOLO) (Long et al., 2020), is the recent advancement based on Paddle Paddle Detector Framework. It employs ResNet50-vd as Backbone, Feature Pyramid Network (FPN) with DropBlock regularization as Neck and YOLOv3 as Head. The overall AP value is increased to 1.5% by introducing Intersection over Union (IoU) Loss and Grid Sensitive modules. Furthermore, the Matrix-based Non-Max Suppression and the CordConv increases mAP value by 0.8%.

The YOLO-based object detection framework outperformed since it detects the objects in a single shot. In this article, the different novel YOLO architectures and their implementations are compared. The metrics AP50, the average precision at IoU of 50, and APM, the average precision with the object size medium are considered for comparison.

In this article, for the Smart Traffic Management System, the YOLOv4 based architecture is employed to detect the vehicles.

Overview of machine learning prediction models

Machine learning-based regression models are categorized into two types: polynomial curve-based approach and decision tree-based approach.

The Elastic Net (Zou & Hastie, 2008) and Support Vector Machine regressor (Steinwart & Christmann, 2008) are based on polynomial approach. While the Random Forest (Breiman, 1996) and eXtreme Gradient Boosting (Chen & Guestrin, 2016) are the decision tree-based approach.

The Elastic Net is based on the Lasso and Ridge regularization methods. The regularization allows to train only the essential features of the model while penalizing the rest and prevent from overfitting. Equation (2), represents the polynomial regression, in which the corresponding constants b are multiplied for each polynomial input x. These values optimized using Gradient Descent Optimizer. Equation (3) represents the Gradient Descent cost function. The yi represents the actual feature value and β′ xi is the predicted value and λ1 and λ2 are the regularization constants.

(2) argminβy=b0+b1x1+b2x12+…+bnx1n

(3) argminβ∑i(yi−2b′xi)2+λ1∑k=1|βk|+λ2∑k=1βk2

In the Support Vector Machine, the polynomial kernel computes the optimal fit hyperplane by scaling down the maximum margin classifier’s error rate within a threshold ε. Thereafter, the hyperplane function is computed by optimizing the primal function (Vapnik, 1995). The support vectors are the data points that are near to the boundary of hyperplane and at ε distance far. Equations (4) and (5) represent the kernel equation of the Support Vector Machine.

(4) y=∑i=1N(αi−αi∗)⋅⟨φ(xi),φ(x)⟩+b

(5) y=∑i=1N(αi−αi∗)⋅K(xi,x)+b

Random forest is a bagging technique based on decision trees, whereas the Gradient Boosting tree (GBT) is an ensemble boosting technique. Bagging is a technique that determines the output by considering all the values of the decision tree. The Information Gain based on Gini Index and Entropy determines the split of the decision tree. The Gini Index and Entropy are the two methods used to calculate the Information Gain, depicted in Eqs. (6) and (7). The boosting technique involves weight adjustment based on the previous decision tree prediction. The residual error Y¯=Y|Fo(x) is determined in both the bagging and boosting techniques. This residual error is added to the initial value and fine-tuned until the value approaches close to the ground truth values. The eXtreme Gradient Boosting (XGBoost) is an advanced Gradient Boosting Tree Algorithm with built-in cross-validation capability, efficient handling of missing data, regularization to avoid overfitting, catch awareness, tree pruning, and parallelized tree building. These are all features that contribute to XGBoost’s robustness. Equation (8) shows the objective function of XGBoost, which is minimized using the second-order Taylor polynomial approximation.

(6) Entropy=∑i=1C−pi∗log2⁡(pi)

(7) GiniIndex=1−∑i=1C(pi)2

(8) L(t)=∑i=1nl(yi,y^i(t−1)+ft(xi))+Ω(ft)

Materials and methodologies

Proposed methodology for smart traffic management system

This section discusses the proposed methodology for Smart Traffic Management System. The proposed method is simple, robust, accurate, and applicable to traffic circles, cross-roads, and fly-overs. Figure 1 presents the framework of the proposed system. The architecture comprises two phases. The first phase involves object detection, and the second phase involves Machine-Learning based regression. Object Detection detects the different vehicle classes, whereas Machine Learning based regression predicts the optimal time of the green light window.

Figure 1 Flowchart for the Smart Traffic Management System.

The initial step of the proposed approach is to fetch the IP address of the CCTV camera from each lane. After that, the region of interest is determined. The Perspective Transformation (Shakunaga & Kaneko, 1989) Algorithm is utilized for determining the region of interest.

The obtained frame is processed by the State-of-the-Art YOLOv4 object detector, which counts the number of cars, buses, trucks, and bikes on the road. Furthermore, the precipitation information is fetched from the Open Weather Map API (OpenWeatherMap, I, 2016) based on the latitude and longitude of the CCTV Camera’s location since precipitation has a significant impact on lane clearance time. Every hour, the precipitation status is retrieved.

The count of different vehicles’ categories and the precipitation information passed as an input to the Machine Learning based Prediction model. This article proposes the eXtreme Gradient Boosting Algorithm (XGBoost) (Chen & Guestrin, 2016), which is fast, efficient, accurate, and prune to overfit. The XGBoost predicts the optimal lane clearance time for the green light window once the assigned previous lane’s timer is completed.

The entire process works seamlessly without creating any buffer zone in processing time. The processing of the next lane is completed in the last 5 s when there is a yellow light. As a result, lane processing and prediction are performed once in a complete sprint. The flow of the whole system is depicted in Algorithm 1.

Algorithm 1 Algorithm for green light prediction.

input :Set of IP Addresses (N)	
Set of Region of Interest (N)	
output :Predicted time of Green Signal	
begin	
LANE ←0;	
TIMER←0;	
Loop	
LANE ← (LANE +1)modN;	
VEHICLES ← YOLOv4.Detect( f rame);	
foreach objects ob j in VEHICLES do COUNTVEHICLES[ob j] ← ob j.value ;	
RAIN ← 0;	
isPrecipitation ← WEATHER API(LAT,LON);	
if isPrecipitation then	
RAIN ← 1;	
PREDICTED TIME ← XGBoost.Predict(COUNTVEHICLES);	
while TIMER not 0 do	
WAIT;	
TIMER←PREDICTED TIME;	
GREEN SIGNAL ← TRUE;	
if TIMER is 5 then	
YELLOW SIGNAL ← TRUE;	

Our algorithm’s main modules are (1) Obtaining the vehicles’ density using the YOLOv4 Object Detection Model and (2) Predicting the optimal time for the green light window using the eXtreme Gradient Boost Prediction Model, which is briefly described in the subsequent sections.

Object detection

The OpenCV-based Leaky YOLOv4 is proposed for analyzing the density of vehicles presented over the lane. The classes that are taken into consideration for vehicles’ density are Car, Heavy Loaded Vehicles, and Bike. Fixed confidence of 25% while 5% for the Tiny model, NMS threshold of 50% are configured as hyper-parameter for Object Detection. After that, the count of vehicles is passed to determine the prediction time.

Prediction model

Polynomial curve fitting and tree-based regression models are proposed to predict the green light window timer. The methodology followed for training the model is to split the dataset into the training part and testing part. Furthermore, the 10-fold cross-validation technique (Browne, 2000) is applied, which divides the training data into the training set and validation set. The model is first trained on training data and then fine-tuned over validation data. Finally, the model’s unbiased evaluation is performed on the unseen test dataset.

Along with the count of vehicles, the precipitation details are fetched from Open Weather Map API (OpenWeatherMap, I, 2016) in one hot encoded form and conveyed to the model.

The predicted time is set to the green light window once the previous lane’s timer is completed. When the last 5 s remain, the light of the current lane switches from green to yellow, and the next lane’s processing is performed.

This process is repeated for each lane. Therefore in our proposed algorithm, the processing of lanes and predictions are performed only once per cycle.

Datasets

The proposed architecture comprises of two models: Object Detection Model and Prediction Model. Hence two distinct datasets are used.The Microsoft Common Objects in Context (MS COCO) dataset (Lin et al., 2014) is considered for object detection. This dataset is often used for performing classification, detection, and segmentation. The dataset consists of 91 different types of objects with around 328K images. The Car, Bus, Truck, and Bike classes are segregated, and the vehicle’s count is passed to the prediction model.

A dataset of 1128 sample points from six different crossroads in Vadodara City, Gujarat, India, is acquired for training the prediction model. These particular crossroads are illustrated in Fig. 2. The total dataset is partitioned into 90% of the training set and 10% of the testing set, which comprises 1,015 as the sample training data points and 113 testing data points. For the validation, the 10-Fold cross-validation (Browne, 2000) technique is utilized, which further split the training dataset into the validation dataset; with the equal partition of 10 blocks.

Figure 2 Cross-Roads undertaken for analytics from Vadodara City—(Map data ©2021 Google).

Results and discussion

The proposed Algorithm is a two-step approach. In the first step, the count of the independent category of vehicles is obtained, which serves as an input to the next step for predicting the optimal time of the green light window. Consequently, the whole approach is the detection using the YOLOv4 (You Only Look Once version4) object detection Algorithm and prediction using the XGBoost (eXtreme Gradient Boosting) Algorithm; As a result, it is an amalgamation of YOLOv4 and XGBoost for regulating the traffic present at the lane.

Analysis of object detection algorithm

The inference time is essential in traffic management; hence, it is reasonable to construct a better accuracy model with lesser inference time. Comparative performance analysis of different YOLO-based detectors is performed to evaluate the accuracy and inference time over CPU in a constrained environment. The six different models are proposed and evaluated including OpenCV DNN based leaky and Mish YOLOv4 (Bradski, 2000), Open Neural Network Exchange (ONNX) based YOLOv4 (Bai et al., 2019), PP-YOLO (Long et al., 2020), Darknet YOLOv4, and Darknet YOLOv4 lite (Bochkovskiy, Wang & Liao, 2020) from MS COCO dataset. A fixed confidence level of 5% is defined for the tiny model in the experiment, and a threshold of 25% is defined for the remaining models. The NMS threshold is set to 50%, with an input image size of 1,359 × 720 × 3 for Day Time and 1,366 × 768 × 3 for Evening Time. The network resolution is set to 416 × 416 × 3 pixels. Figs. 3 and 4, depict the study of the different object detection algorithms during the day and at night under low—light conditions.

Figure 3 Different object detection algorithm analysis for traffic at cross-road in the day time (A) OpenCV DNN Leaky YOLOv4 (B) OpenCV DNN Mish YOLOv4 (C) ONNX YOLOv4 (D) PP-YOLO (E) Darknet YOLOv4 (F) Darknet YOLOv4 Tiny.

Figure 4 Different object detection algorithm analysis for traffic at cross-road in the night time (A) OpenCV DNN Leaky YOLOv4 (B) OpenCV DNN Mish YOLOv4 (C) ONNX YOLOv4 (D) PP-YOLO (E) Darknet YOLOv4 (F) Darknet YOLOv4 Tiny.

The inference and mAP analysis is described in Table 1. The accuracy and inference time of object detection models are evaluated under the constrained environment with 1 × single core hyperthreaded Xeon Processors @2.3GHz (1 core, 2 threads) with 12.6 GB RAM. From this analysis, it is concluded that lite models are remarkably fast with 0.44 s in day time and 0.1 s in evening time as compared with other models as shown in table, but the error rate is high, resulting in a significant impact over predicting the optimal time. Darknet YOLOv4 is highly with 63.3% of AP50 as compared with other models, but the inference time exceeds 5 s, making it infeasible. Although the PP-YOLO using conventional greedy NMS is more accurate with 45.2% AP and faster than Darknet YOLOv4, its inference time is closer to 5 s. The AP value of YOLOv4 ONNX and OpenCV DNN Mish is the same; however, since OpenCV is substantially optimized for CPU, OpenCV DNN has a lesser inference time than ONNX. Equations (9) and (10) represent the Mish Activation Function and Leaky Activation Function respectively.

Table 1 YOLO object detection comparison between YOLOv4 ONNX, YOLOv4 Darknet, YOLOv4 Darknet Tiny YOLOv4, PP-YOLO, OpenCV Leaky YOLOv4 and OpenCV YOLOv4.

The inference time and accuracy is calculated by using fixed computational environment.

Architecture	Inference time
in Day light	Inference time
in Evening
(low light)	AP50
(416 × 416)	APM
(416 × 416)	
YOLOv4 ONNX	∼3.1327 s	∼3.168 s	63.3%	44.4% AP	
YOLOv4 Darknet	∼8.865 s	∼8.965 s	63.3%	44.4% AP	
YOLOv4 Darknet Tiny	∼0.44 s	∼0.1 s	40.2%	–	
PP-YOLO	∼4.489 s	∼4.468 s	62.8%	45.2% AP	
OpenCV Leaky YOLOv4	∼1.40821 s	∼1.4109 s	62.7%	43.7% AP	
OpenCV Mish YOLOv4	∼1.6733 s	∼1.679 s	63.3%	44.4% AP	

(9) f(x)=x⋅tanh⁡(softplus(x))=x⋅tanh⁡(ln⁡(1+ex))

(10) LeakyReLUmax(0.1x,x)

In YOLOv4, a combination of Mish function and Cross Stage Partial Network (CSPDarknet53) is utilized, which, although slightly expensive in inference time but significantly improves detection accuracy. The Mish and Leaky YOLOv4 are quite close. As a result, OpenCV-based Leaky YOLOv4, having optimal inference time and accuracy as shown in table, is proposed for smart traffic management system (Bradski, 2000).

Analysis of prediction model

Elastic Net, Support Vector Regressor, Random Forest Regressor, and Extreme Gradient Boosting algorithms are analyzed to determine the prediction model. Elastic Net and Support Vector Regressors are polynomial-based regressors, while Random Forest and Extreme Gradient Boosting are decision tree-based regressors. Table 2 details the fine-tuned hyper-parameters and regression analysis of each model across the training, validation, and test datasets. The Bayesian parameter optimization technique from the hyperparameter framework “Optuna” is considered for parameter optimization and function subset selection (Akiba et al., 2019). The r-squared coefficient of determination and the mean squared error (MSE) are used as an evaluation metrics, depicted in Eqs. (11), (12) and (13). Based on the analysis findings, it is concluded that the decision tree-based models are best fit than polynomial-based models. Furthermore, studies shows that Elastic Net performed poorly with a R2 score of 0.638 and an MSE value of 21.79 over unseen samples, whereas Extreme Gradient Boosting outperformed and shows promising results with a R2 score of 0.92 and an MSE value of 5.53 in the test set. Furthermore, when compared to other models, XGBoost R2 score is greatest in the training and cross validation sets, with 0.9944 and 0.9384, respectively, and it has the lowest Mean Square Error of 1.113 and 11.576, as shown in Table 2.

Table 2 Regression Model Comparison between Elastic Net, Support Vector Machine Regressor (SVR), Random Forest Regressor and eXtreme Gradient Boosting Tree based (XGBoost GBT).

The hyperparameters of each model are optimized for training, validation and testing set.

Model	Hyper-parameters	Training set	Cross validation	Testing set	
Elastic Net	Degree: 2	R2: 0.7891	R2: 0.7936	R2: 0.638	
	Interaction: True	MSE: 34.56	MSE: 36.672	MSE: 21.79	
	Learning Rate: 0.05				
	L1 ratio: 0.5				
SVR	Kernel: poly	R2: 0.8207	R2: 0.7964	R2: 0.7321	
	Degree: 3	MSE: 35.157	MSE: 35.323	MSE: 22.39	
	C: 4.688				
Random Forest regressor	N estimators: 120	R2: 0.98	R2: 0.8932	R2: 0.8933	
	Max depth: 58	MSE: 3.592	MSE: 18.918	MSE: 19.16	
XGBoost GBT	N estimators: 76	R2: 0.9944	R2: 0.9384	R2: 0.92179	
	max depth: 120	MSE: 1.113	MSE: 11.576	MSE: 5.513	
	learning rate: 0.35				
	gamma: 0.018				
	base score: 0.578				

(11) r2=1−SumofSquareofResidualError(ssres)SumofSuareofTotalVariation(sstot)

(12) SSres=∑i((yi)−fi)2=∑iei2andSStot=∑i((yi)−(¯y))2

(13) MSE=1n∑i=1n(yi−yi¯)2

The randomly selected data points from the testing dataset are analyzed in Table 3. The XGBoost prediction model considers the number of vehicles and precipitation details to determine the optimal time. Thereafter, the predicted time is compared with the static time, and the calculation of reduced waiting time is performed, which is depicted in Eq. (14). The reduced average waiting time is 32.3% in comparision with the static time allocated to the usual road traffic based on the test dataset. Hence the proposed methodology reduces the intersection’s waiting time and improves the traffic lights’ efficiency.

Table 3 Waiting Time Reduced by XGBoost in comparison to Static Time.

The parameters used for prediction are number of Cars, Bus or Trucks, Bike and Precipitation.

CAR	BUS and TRUCK	BIKE	RAIN	XGBoost predicted (in s)	Static Time (in s)	Reduced waiting time	
1	6	5	YES	48	60	20%	
4	5	2	NO	48	60	20%	
5	2	5	YES	40	60	33%	
7	1	3	YES	44	60	27%	
2	4	5	YES	35	60	42%	
3	2	4	NO	32	60	47%	
4	1	1	YES	27	60	55%	

(14) 1N∑i=1N(Originaltime−Predictedtime)Originaltime∗100%

The histogram along with kernel density estimation (KDE) for prediction model is depicted in Fig. 5. The data distribution is based on the density of vehicles against the time predicted by the model. The probability density of continuous or non-parametric data variables is visualised using the Kernel Density Estimate. It is calculated as the region in the interval between the density function (graph) and the x-axis which is represented in the Eq. (15). The b is representing the bandwidth of bin, the kernel density function K(.) is the chosen Kernel weight function and xi is the observing data point. The KDE reflects how far the individual data point is from the mean data point in the same bin. Furthermore, this plot clearly shows that the distribution is not skewed. The average predicted value under normal traffic is 46.834 s while the median value is 45.01 s. These indicates that their is a symmetric distribution in predicted time with respect to density of traffic. The Minimum predicted time is 14.98 s and the maximum time predicted is 100.03 s.

Figure 5 Histogram along with Kernel Density Estimation for the predictive model.

(15) f^Kernel(x)=1Nb∑i=1NK(xi−xb)

Figure 6 highlights the variation of time based on the density of vehicles. The considered range is from 0% to 100% with an interval of 25% and the allocated times are 15 to 40, 25 to 60, 40 to 90, and 75 to 100 s for each interval. The average time predicted in 25% density is 35.84 s, for 50% density is 45.017 s, for 75% density is 54.91 s and 87.32 s in 75% to 100% density. The average difference between labelled time and model prediction is minimal, indicating that the model has a perfect fit over the Gaussian probability density curve. The comparison with the actual allocated static time over the Gaussian probability distribution is presented in Fig. 7. This figure demonstrates that the proposed solution effectively decreases waiting time as opposed to the static allocated time.

Figure 6 Average value time between model predicted and labelled for training (A) 25% of Vehicles (B) 50% of Vehicles (C) 75% of Vehicles (D) 100% of Vehicles.

Figure 7 Average value time.

Conclusion

In this work, a unified algorithm for the smart traffic management system is introduced to address traffic congestion by predicting the optimal time of the green light window. The proposed approach replaces the conventional method of allocating timer in a round-robin pattern with the dynamic time allocation. This research includes the use of incoming frames from the CCTV cameras. As a result, integrating it with traffic lights is simpler and efficient. The proposed approach is scalable as the system works appropriately during the high stream and in low traffic and it is cost-efficient as it does not require much maintenance or any sensors; instead, it is completely based on input frames from the CCTV cameras.

Furthermore, the proposed approach is robust, versatile, and works efficiently during the day and at night time with low street lights, and it works flawlessly under all weather conditions. An additional monitoring system is provided to manage the failure, in which the notification sent to the control room, and traffic lights can be handled manually. The real-time traffic data is used and gives predictions based on the independent category of vehicles present on the lane. As a result, the unnecessary waiting time and fuel consumption are reduced, resulting in lower air pollution. As the article modernizes and renovates the conventional methodology of Traffic Management System with an Artificial Intelligence-based approach, it is theorized as a Smart Traffic Management System.

The future scope is to implement the lane clearance for an emergency vehicle, assessing the weather by classifying the frame, considering specific time events such as rush hour and peak hours into the effect.

Supplemental Information

Supplemental Information 1 Code of Smart Traffic Management System.

The YOLO object detection analysis and prediction models are in the Jupyter-Notebook file. The main script which starts the system and other files referenced with the main script is available in the Python file and the whole system is implemented in python programming language.

The pretrained model of YOLOv4 is available at GitHub: https://github.com/AlexeyAB/darknet/wiki/YOLOv4-model-zoo.

Click here for additional data file.

Supplemental Information 2 Dataset of Smart Traffic Management System.

The information of features that are count of vehicles and precipitation details along with the output value of prediction time.

Click here for additional data file.

The Vadodara Municipal Corporation provided the video data from six different crossroads of Vadodara City to perform analysis and research. The Charotar University of Science and Technology (CHARUSAT) provided the platform and hardware resources to perform comprehensive experiments for the research.

Additional Information and Declarations

Competing Interests

Author Contributions

Data Availability

The authors declare that they have no competing interests.

Pritul Dave conceived and designed the experiments, performed the experiments, performed the computation work, prepared figures and/or tables, authored or reviewed drafts of the paper, and approved the final draft.

Arjun Chandarana conceived and designed the experiments, performed the experiments, performed the computation work, prepared figures and/or tables, authored or reviewed drafts of the paper, and approved the final draft.

Parth Goel conceived and designed the experiments, analyzed the data, authored or reviewed drafts of the paper, and approved the final draft.

Amit Ganatra conceived and designed the experiments, analyzed the data, authored or reviewed drafts of the paper, and approved the final draft.

The following information was supplied regarding data availability:

The dataset and script are available in the Supplemental Files.

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
