# Peer review of "An amalgamation of YOLOv4 and XGBoost for next-gen smart traffic management system"

_PeerJ Computer Science, doi:10.7717/peerj-cs.586_

## Round 0.1 · original submission · Major Revisions

Please revise the manuscript based on the reviewers' comments and provide detailed responses to the raised concerns and comments.

Reviewer 1 ·

Basic reporting

The review of the state of the art is insufficiently addressed. Literature references should be improved. See for example:
F. Castaño, G. Beruvides, R. E. Haber, and A. Artuñedo, “Obstacle recognition based on machine learning for on-chip lidar sensors in a cyber-physical system,” Sensors (Switzerland), vol. 17, no. 9, 2017, doi: 10.3390/s17092109.
F. Castaño, G. Beruvides, A. Villalonga, and R. E. Haber, “Self-tuning method for increased obstacle detection reliability based on internet of things LiDAR sensor models,” Sensors (Switzerland), vol. 18, no. 5, 2018, doi: 10.3390/s18051508.
The quality of figures should be improved. Font size is too small in some cases.

Experimental design

Other machine learning methods should be explored. Why are not explored self-parameterization using gradient-free optimization methods? See for example:
R.-E. Precup and R.-C. David, Nature-Inspired Optimization Algorithms for Fuzzy Controlled Servo Systems, Butterworth-Heinemann, Elsevier, Oxford, UK, 2019.
R. Haber et al., "A Simple Multi-Objective Optimization Based on the Cross-Entropy Method," IEEE Access, vol. 5, pp. 22272-22281, 2017.
G. Wang and L. Guo, "A novel hybrid bat algorithm with harmony search for global numerical optimization," Journal of Applied Mathematics, vol. 2013, 2013.
R. H. Guerra et al., “Digital Twin-Based Optimization for Ultraprecision Motion Systems with Backlash and Friction,” IEEE Access, vol. 7, pp. 93462–93472, 2019, doi: 10.1109/ACCESS.2019.2928141.

Validity of the findings

Other performance indices and techniques should be included in the comparison to assess the real contribution of the proposed approach.
Conclusions are adequate according to the study.

Additional comments

The work addresses an interesting topic however the novelty and the progress beyond the state of the art is not sufficiently outlined.

Reviewer 2 ·

Basic reporting

-This paper does not present a Professional English, since the authors must write the acronyms correctly, and some acronyms do not have their meaning. Correct this error. Also, the words "Figure", "Table", "Equation", "Algorithm", must be written with the first letter in capital letters. Review and correct it in the article.
-The authors have not placed some reference scientific literatures, so I suggest performing a more in-depth investigation and adding these articles:
----- Zambrano-Martinez, J. L., Calafate, C. T., Soler, D., Lemus-Zúñiga, L. G., Cano, J. C., Manzoni, P., & Gayraud, T. (2019). A Centralized Route-Management Solution for Autonomous Vehicles in Urban Areas. Electronics, 8 (7), 722.
--- Zhang, X .; Onieva, E .; Perallos, A .; Osaba, E .; Lee, V. Hierarchical fuzzy rule-based system optimized with genetic algorithms for short term traffic congestion prediction. Transp. Res. C Emerg. Technol. 2014, 43, 127–142
-The structure of the article is well detailed and will match the results with the hypothesis they present.
-The results are a bit weak on the part of the authors, because the authors must include some Figure with their respective explanation, when the scenario is not applied the prediction, and when the scenario is applied the prediction, that is noticed when it has exceeded the prediction.

Experimental design

-The authors do not present which scenario is being studied. Furthermore, they do not present the parameters for that study. Some Figure of the stage must be placed.
-Figure 2 can be transformed into an Algorithm, if it is done it would give more quality to the paper.
-Is there a figure that shows the prediction that the authors comment?
-With which simulator has the study been performed?
-More graphics are needed for the reader to understand more about the work done.
-How is the Smart Traffic Management System presented by the authors composed?
-What are the routes that the vehicles have taken to predict traffic?

Validity of the findings

-The authors do not present where the data comes from to perform the experiment. Which must be detailed, and referenced
-The conclusions must be improved according to the new implementations suggested by the reviewers that support the results.

Additional comments

-The authors must perform the changes suggested by the reviewers to improve the quality of the article.

---

## Round 0.2 · Minor Revisions

Please further address the reviewer's comments.

Reviewer 1 ·

Basic reporting

English should be carefully checked.
The review of the state of the art is not sufficiently addressed according to reviewers' suggestions.

Experimental design

Experimental design is fine.

Validity of the findings

Novelty is adequately addressed.

Additional comments

-

---

## Round 0.3 · Minor Revisions

Please conduct the revisions as requested by the comments from the reviewers.

Reviewer 1 ·

Basic reporting

The paper can be accepted.

Experimental design

The paper can be accepted.

Validity of the findings

The paper can be accepted.

Additional comments

The paper can be accepted.

Reviewer 2 ·

Basic reporting

-I suggest adding more references such as:
--Zambrano-Martinez, J. L., Calafate, C. T., Soler, D., Lemus-Zúñiga, L. G., Cano, J. C., Manzoni, P., & Gayraud, T. (2019). A centralized route-management solution for autonomous vehicles in urban areas. Electronics, 8(7), 722.
--Lv, Y.; Duan, Y.; Kang, W.; Li, Z.; Wang, F.Y. Traffic flow prediction with big data: A deep learning approach.
IEEE Trans. Intell. Transp. 2014, 16, 865–873

Experimental design

--There are no Figures of which are the polynomial curve for the predictive model.

Validity of the findings

Everything is fine.

Additional comments

Perform the suggested modifications.

---

## Round 0.4 · accepted · Accept

The authors have addressed all the comments from the reviewers.